# Estimating ecoacoustic activity in the Amazon rainforest through Information Theory quantifiers

**Juan G. Colonna** **[1]***, **José R. H. Carvalho[1], Osvaldo A. Rosso[2]**

**1** Instituto de Computação (IComp), Universidade Federal do Amazonas (UFAM), Manaus, Amazonas, Brasil, **2** Instituto de Física, Universidade Federal de Alagoas (UFAL), Maceió, Alagoas, Brasil

* juancolonna@icomp.ufam.edu.br

## Abstract

Automatic monitoring of biodiversity by acoustic sensors has become an indispensable tool to assess environmental stress at an early stage. Due to the difficulty in recognizing the Amazon's high acoustic diversity and the large amounts of raw audio data recorded by the sensors, the labeling and manual inspection of this data is not feasible. Therefore, we propose an ecoacoustic index that allows us to quantify the complexity of an audio segment and correlate this measure with the biodiversity of the soundscape. The approach uses unsupervised methods to avoid the problem of labeling each species individually. The proposed index, named the Ecoacoustic Global Complexity Index (EGCI), makes use of Entropy, Divergence and Statistical Complexity. A distinguishing feature of this index is the mapping of each audio segment, including those of varied lengths, as a single point in a 2D-plane, supporting us in understanding the ecoacoustic dynamics of the rainforest. The main results show a regularity in the ecoacoustic richness of a floodplain, considering different temporal granularities, be it between hours of the day or between consecutive days of the monitoring program. We observed that this regularity does a good job of characterizing the soundscape of the environmental protection area of Mamirauá, in the Amazon, differentiating between species richness and environmental phenomena.

## Introduction

It is well established that animal species are sensitive to their environmental conditions. In addition, recent research shows that climate change modifies the Earth's natural soundscapes [1]. These two observations have driven many researchers to monitor the variations of animal populations through time using acoustic measures and indices, and to use them as indicators of environmental degradation [2]. Currently, there are several ways to achieve this goal, and two of the most employed are: by acoustic surveys or by acoustic diversity indices. Acoustics surveys are the most widely used method for monitoring animal populations, taking advantage of the animal vocalization capability [3]. However, a solid survey in remote tropical areas, such as the Amazon rainforest, demands a significant investment of both human and financial

**Data Availability Statement:** The data belong to the Providence project, which is a research partnership of four institutions. For this reason, raw data (without preprocessing) must be requested on the website (http://www.projectprovidence.org) or

through the institutional system SISIBio (https://www.icmbio.gov.br/sisbio/). We made this request and we obtained the registration number 72722 (SISBio). Therefore, the same data can be requested for research purposes. However, in Zenodo (https://doi.org/10.5281/zenodo.3866662), we have provided a link with a preprocessed copy of the dataset in which we included the main acoustic indices already calculated. The dataset was made available in dataframe format with the features extracted from the raw audio data. Besides this dataset, the entry also contains all the codes used in the experiments. In this way, we allow the transparent replication of our results with less effort for anyone who is interested.

**Funding:** The author(s) received no specific funding for this work.

**Competing interests:** The authors have declared that no competing interests exist.

resources as well as expert knowledge. Diversity indices, on the other hand, belongs to a broader class of methods capable of quantifying diversity without needing to recognize each particular sound or the species that produce it [4–6].

Currently, most ecoacoustic recognition methods rely on supervised classifiers to automatically identify and catalog species. Researchers apply these methods to a variety of species, including frogs, birds, whales, dolphins, elephants, mosquitoes, gibbons, among others [7–13]. Fully automatic multi-species monitoring systems have also been proposed, integrating hardware and software on a single platform [14]. Although the ultimate goal of supervised methods is to monitor specific species, these methods can also be used as indicators of soundscape changes due to variability in the detection of monitored species [15, 16]. However, all of these works are faced the same challenge: the manual labeling of large databases of long audio recordings by a collaborator with expert knowledge. Furthermore, a well known shortcoming of the supervised methods is the decrease in classification performance as the number of classes increases. [17].

An alternative way to deal with these issues is through the use of unsupervised methods [16]. Clustering methods do not require labeled data and are particularly well suited in analyzing the acoustic diversity within given soundscapes, such as a tropical environment [18]. However, determining the number of clusters and validating the content of each recording group not only requires expertise but is also time consuming and subjective. Moreover, the fact that clusters can be heterogeneous in distinctive ways transforms assigning biodiversity scores to each group into a substantial task.

Another class of unsupervised methods has emerged within the last decade. Most of them define an acoustic index, usually linked to the acoustic richness of the audio recordings. Methods based on acoustic indices are particularly suited for ecoacoustic data analysis. Some examples of such indices are the Vocal Activity Index (VAI) [16], the Acoustic Complexity Indices (ACI) [6, 19, 20], the Bioacoustic Index (BI) [21], the Acoustic Entropy Index ($H_a$) [4], the Acoustic Diversity Index (ADI) [22], the Acoustics Evenness Index (AEI) [22], and the Normalized Difference Soundscape Index (NDSI) [23, 24]. However, there is no consensus about the acoustic index that best describes a landscape configuration [25]. As pointed out by Fairbrass *et. al.* [26], most of the mentioned indices are biased due to their positive correlation with anthropogenic or geophonic noises.

The difficulty of having a non homogeneous characterization of a soundscape using different acoustic indices is that most of them are altered by environmental noise in the recordings [25]. It is known that any real signal recorded in the field, $X$ is formed by the composition of two components ($X = \mathcal{X} + \mathcal{N}$) [27, 28]. Being $\mathcal{X}$, the deterministic component, possibly caused by animals with regular singing patterns or vocalizations, and the stochastic component $\mathcal{N}$, commonly labeled as background noise. This composition might be seen in the spectrograms of Fig 4d. Consequently, two records with similar acoustic patterns but distinct background noise can lead to differing ecoacoustic characterizations. This is particularly true for indices based only on Shannon's entropy.

Several indices have been defined based on Shannon's entropy [4, 22, 29]. There is also a consensus in the ecoacoustic literature to consider entropy as a synonym for complexity; therefore, the higher the entropy, the higher the complexity [24]. Entropy is a global attribute capable of quantifying uncertainty. When used to characterize a physical system, entropy describes a state's disorder, not its complexity [30]. A classic example of this dilemma is an isolated ideal gas system, in which the behavior of gas particles tends to balance with maximum entropy ($H \rightarrow 1$). Since the state of the system itself is quite simple and easy to describe, complexity should be minimal ($C \rightarrow 0$) [31]. For a richer discussion on the

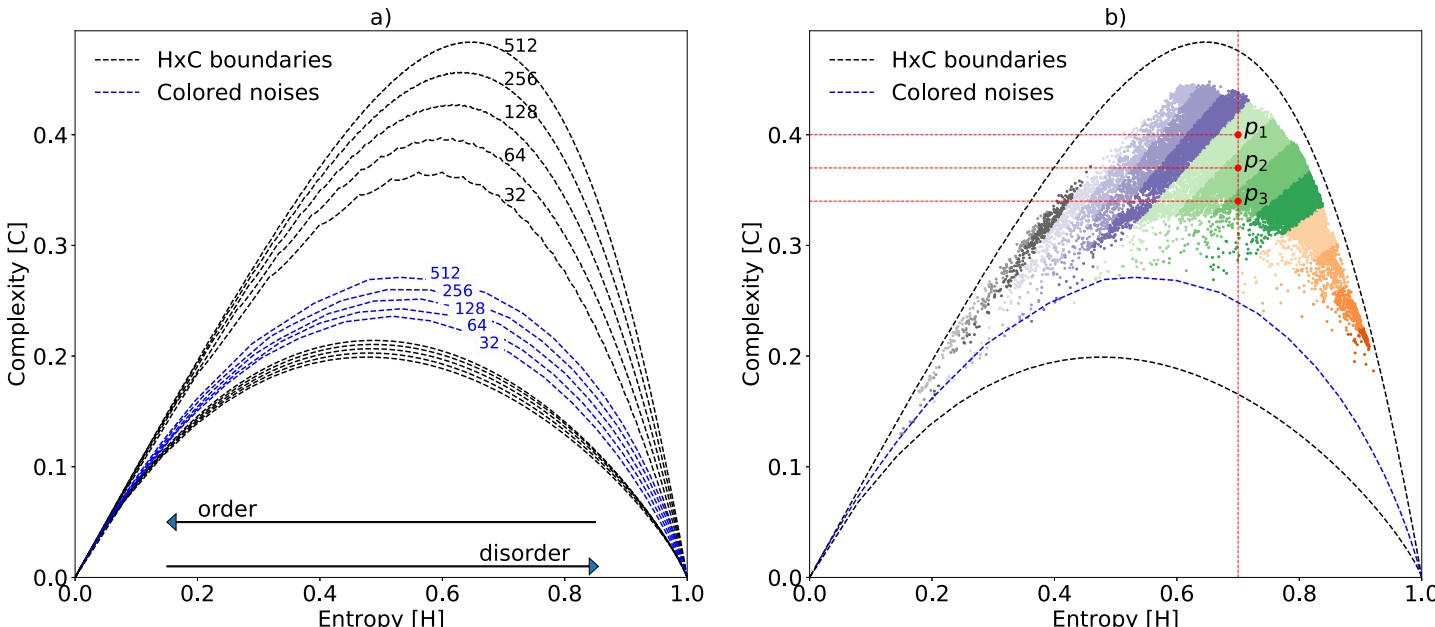

**Fig 1. Boundaries of the HxC-plane and the color noises reference line.** (a) Maximum and minimum boundaries of the Generalized Statistical Complexity represented according to the chosen time lag $\tau_{max}$. Here, the blue dashed line shows the variation of the correlated noises according to $\alpha$. (b) Distribution of some acoustic samples from our dataset when $\tau = 512$. Points $p_1$, $p_2$ and $p_3$ have a singular spectrum with equal entropy but different complexity. The color gradient illustrates regions of the plane in which the samples have similar divergence.

differences between entropy and complexity, please refer to the seminal articles on this topic [28, 30, 31].

This article summarizes our effort to bring these concepts to the ecoacoustic field. By analogy, we can consider the soundscape as our physical system, where recordings are the system's measurable outputs. Thus, by defining an appropriate methodology to represent the probabilities of the states from the outputs, we can quantify the complexity of the soundscape. We adopted the definition of complexity illustrated in Fig 1a, where the complexity is zero for both states of the system: totally ordered and totally disordered [32]. Ordered systems are those having predictable acoustic patterns ($H \rightarrow 0$ and $C \rightarrow 0$), and disordered systems are those with only white noise ($H \rightarrow 1$ and $C \rightarrow 0$), or totally unpredictable acoustic patterns.

The contributions of this work are two-fold: a new methodology for processing ecoacoustic signals that allows us to obtain a measure of complexity and represent any record as a unique two-dimensional point, and a detailed data analysis using a set of real recordings from an acoustic monitoring program applied in an environmental preservation area. The proposed index, named the Ecoacoustic Global Complexity Index (EGCI), is able to track the ecoacoustic status of a given landscape. The index promotes a better understanding of variations in the soundscape caused by animal vocalizations and environmental phenomena. The EGCI is unsupervised, meaning that no human expertise is needed to label and classify long audio records. The EGCI benefits from three Information Theory quantifiers: the Shannon entropy, the Jensen-Shannon divergence, and the Generalized Statistical Complexity, which, when combined, are able to differentiate between a broad range of signal features, in both temporal and frequency domains.

The EGCI is derived from the Entropy-Complexity HxC-plane, a well-established technique in physics used to study the dynamics of complex systems [28, 33]. With this plane, it is possible to characterize samples from a specific soundscape. We also present a detailed data

analysis on a set of continuous recordings, generated by a sensor node positioned at a preserved floodplain area known as the *Mamirauá Reserve*, located in the central Amazon rainforest in Brazil. Therefore, if we consider a soundscape as a complex physical system, the EGCI enables us to study and interpret the acoustic dynamics of that system not only for a specific set of species, but as a whole.

## Related works

The existing acoustic indices can be categorized into two main categories: the Alpha and the Beta indices [24]. Alpha indices were developed in order to quantify the acoustic richness of a given soundscape while the Beta indices were proposed as a way to asses the level of acoustic disparity between recordings of different soundscapes. Here we focus on the Alpha indices.

The authors of the Vocal Activity Index (VAI) proposed training a binary classifier to predict whenever there is a bird call in a recording segment [16]. After that, the number of segments predicted as the positive class is divided by the total segments recorded over the same time period. Then, the results obtained by the supervised classifier are used to validate and estimate this index employing an unsupervised cluster method. All of these processing steps, along with manually adjusting the optimal number of clusters, make it difficult to analyze large data sets or to deploy this method in a low-cost acoustic sensor.

One of the Alpha indices most frequently found in literature is the Acoustic Complexity Indices (ACI) [19]. This index is based on the rate of change provided by each frequency band of the spectrogram, where the spectrogram can be obtained with the Short-Time Fourier Transform (STFT) algorithm. This algorithm makes this index simple and computationally fast. Modern hardware can perform STFT without any difficulty. However, the number of rows and columns of the spectrogram changes as we change both the STFT resolution and the signal length. This mutual dependence between temporal- and spectral-resolution makes it difficult to compare results when the sampling frequency changes, due to hardware updates, or when the temporal length of recordings varies. Furthermore, this index was also used as a feature for a supervised classification method [20]. In order to attain more details on the calculation of this index please refer to S1 Appendix in S1 File.

The Bioacoustic Index (BI) is obtained through the integration of the sound intensity curve (measured in dB) varying between the frequency bands from 2 kHz to 8 kHz [21]. Again, frequency spectra can be obtained by using FFT, and a minimum threshold can be used to filter out background noise. With BI, it is possible to quantify the signal energy accumulated at different times of the day, when there is higher activity of the birds' singing. However, the integral of the intensity curve of a frequency spectrum with high energy peaks in narrow frequency bands could have the same integral of a spectrum with uniformly distributed energy. This makes the distance between the sound source and the microphone an essential factor in the quantification of diversity by BI.

The Acoustic Diversity Index (ADI) uses Shannon's entropy to calculate the dispersion of spectral energy accumulated over time and spread in ten frequency bands equally distributed between 0 kHz and 10 kHz [22, 34]. Four steps must be performed to obtain the vector of probabilities necessary to calculate the entropy: a) generate the spectrogram and normalize it in order to reach a maximum value of zero, b) apply the dB logarithmic transformation relative to full scale, c) binarize the spectrogram by thresholding, which then causes the values below -50 dB to become 0's and the values above -50 dB to become 1's, and d) to add the values in each band in order to obtain the discrete energy accumulated over time. Additional normalization can be used to transform the accumulated values into a Probability Mass Function (PMF). The two key points that differentiate this index from others are the use of binarization, which acts as

a filter for environmental noise or distant sounds, and the discretization of frequency bands. In comparison, EGCI achieves a noise filtering effect through naturally selecting the most representative eigenvalues (see Methods section). Furthermore, it was demonstrated by Fairbrass *et. al.* [26] that ADI does not positively correlate with biotic acoustic activity or diversity.

The Acoustic Evenness Index (AEI) is the Gini coefficient applied to the frequency bands (or PMF) obtained using the same methodology as the ADI [22]. This index measures the distribution of energy accumulated over time between ten frequency bands, and it can vary from 0 to 1, with 0 representing a perfect equal distribution and 1 representing perfect inequality.

The purpose of the Normalized Difference Soundscape Index (NDSI) is to estimate the disturbance caused by the anthropogenic noise in a given soundscape [23]. Similar to ADI, the FFT is applied to obtain the signal's power spectrum discretized into 10 frequency bands. The bands between 2 kHz and 8 kHz are considered biophonic, i.e. where there is a higher bioacoustic concentration. Among them, the band with the highest Power Spectral Density (PSD) is chosen, and this value is then used to compose the $\alpha$ coefficient. The remaining frequency bands, ranging from 0 kHz to 2 kHz and from 8 kHz to 10 kHz, are considered anthropogenic noise. The highest PSD value between these secondary bands is chosen to form the $\beta$ coefficient. Finally, the NDSI estimate is obtained using the quotient $(\beta - \alpha)/(\beta + \alpha)$, where NDSI = 1 indicates a signal that does not contain anthrophony disturbance. Although this index is capable of measuring the level of disturbance caused by humans in the soundscape, it does not appropriately quantify the acoustic diversity, since it considers a single biophonic frequency band.

Recently, Sueur et al. [4] proposed multiplying the Spectral Entropy ($H_f$) and the Temporal Entropy ($H_t$) to compute the Acoustic Entropy Index ($H_a$). To calculate $H_f$ it is assumed that the normalized frequency spectrum, obtained by the Fast Fourier Transform (FFT), is a histogram. Similarly, one should assume that oscillations in the amplitude envelope of the signal can be used as a PMF to obtain $H_t$. To see the calculation details of these entropies please refer to S1 Appendix in S1 File. Then, the final index is the product $H_a = H_f \times H_t$, so the closer $H_a$ to one, the greater the acoustic diversity. One may notice that short-term impulsive noises can drastically alter the value of this index. Unfortunately, the multiplication of two entropy values lacks physical interpretation. However, this index captures global signal properties, and therefore inspired our proposal, as we believe that Information Theory quantifiers are natural choices for quantifying diversity.

We identified three issues when using the indices mentioned above. First, the increase in quasi-white random noise spreads energy evenly across the spectrum, thus increasing the level of entropy. Hence, entropy values close to 1 cannot always be interpreted as high acoustic diversity produced by vocalization patterns from different species. Second, as they are affected by STFT's multi-resolution dilemma, they depend directly on the length of the recorded audio and the number of selected frequency bands. Therefore, signals with variable lengths change the index ranges, making the values no longer comparable. Third, as these indices are mathematically different, in practice they capture different characteristics of the acoustic signals. Depending on the purpose of the monitoring program, these differences can become complementary or competing. The proposed EGCI is less affected by these issues, representing a clear improvement in the subject of measuring ecoacoustic complexity. The benefits of EGCI are explained in the Result section.

## Methods

In this section we present the three fundamental concepts used to calculate the proposed index: (a) the autocorrelation matrix, (b) the *Von Neumann* entropy, and (c) the Statistical Complexity measure.

## Autocorrelation matrix

The Pearson's correlation coefficient ($r_{xy}$) is a measure of the intensity and direction of the linear relationship between two signals $X$ and $Y$. The autocorrelation coefficient $r_{xx}$ has a similar interpretation, but instead of using two different signals, it uses a version of the same signal shifted by $\tau$ units. For instance, if $\tau = 1$ then $r_{xx}$ quantifies the strength of the association between $x_i$ and $x_{i+1}$ [35].

Let $X = \{x_1, x_2, \ldots x_N\}$ be the acoustic signal of length $N$ at the sensor input; the unbiased autocorrelation coefficient $r_{xx}$ is defined as:

$$r_{xx}(\tau) = \frac{1}{(N-\tau)s^2} \sum_{i=1}^{N-\tau} (x_i - \bar{x})(x_{i+\tau} - \bar{x}), \tag{1}$$

where $N$, $\bar{x}$, and $s$ are the length of $x$, the sample mean, and the sample standard deviation, respectively. Here, the maximum value of $\tau$ must satisfy the condition $\tau_{max} \ll \frac{N}{2}$. This equation may also be referred to as the Autocorrelation Function (ACF).

Given a maximum $\tau_{max}$ value, the autocorrelation matrix $R_{xx}$ can be formed as a Toeplitz matrix, or diagonal-constant matrix, with shape [36]:

$$R_{xx}(\tau) = \begin{bmatrix} 1 & r_{xx}(1) & r_{xx}(2) & \cdots & r_{xx}(\tau_{max}) \\ r_{xx}(1) & 1 & r_{xx}(1) & \cdots & r_{xx}(\tau_{max} - 1) \\ r_{xx}(2) & r_{xx}(1) & 1 & \cdots & \vdots \\ \vdots & \vdots & \vdots & \ddots & r_{xx}(1) \\ r_{xx}(\tau_{max}) & r_{xx}(\tau - 1) & r_{xx}(\tau - 2) & \cdots & 1 \end{bmatrix}, \tag{2}$$

where $r_{xx}(0) = 1$.

These autocorrelation coefficients are efficiently calculated by the Fast Fourier Transform (FFT) algorithm. Regardless of how they are obtained, it is well known that these coefficients carry information about the signal's main frequencies. Therefore, we can consider them spectral features that accurately describe ecoacoustic signals. For more details on this, please refer to S2 Appendix in S1 File.

## Entropy methodology

The Von Neumann entropy was defined by 1927 for quantum measurement processes [37]. It has a fundamental role in studying correlated systems. This Information Theory quantifier is defined as the normalized Shannon entropy ($H$) of the singular spectrum as:

$$H[P] = \frac{-1}{\log(\tau_{max})} \sum_{i=1}^{\tau_{max}} \left( \frac{\lambda_i}{\sum_i^{\tau_{max}} \lambda_i} \right) \log \left( \frac{\lambda_i}{\sum_i^{\tau_{max}} \lambda_i} \right), \tag{3}$$

where $\lambda_i$ are the eigenvalues of a given $R_{xx}$ matrix. It is worth mentioning that, the denominator term $\sum_i^{\tau_{max}} \lambda_i$ normalizes the eigenvalues values between $0 \leq \lambda_i \leq 1$. Hence, the whole term $\frac{\lambda_i}{\sum_i^{\tau_{max}} \lambda_i}$ can be interpreted as a histogram. Lastly, the term $\frac{1}{\log(\tau_{max})}$ normalizes the entropy within $0 \leq H[P] \leq 1$.

## Generalized statistical complexity measure

The original proposal of López-Ruiz *et al.* [31] and the extended work of Rosso *et al.* [32] define the Generalized Statistical Complexity Measure as the functional product:

$$C[P] = Q[P, P_e]H[P] \qquad (4)$$

where $H[P]$ is the normalized entropy (Eq 3), $P$ is a normalized histogram obtained from the eigenvalues of $R_{xx}$, $P_e$ is a reference histogram with uniform distribution, and the disequilibrium $Q[P, P_e]$ is defined in terms of the Jensen-Shannon divergence $J[P, P_e]$. That is:

$$Q[P, P_e] = Q_0 J[P, P_e] \qquad (5)$$

with:

$$J[P, P_e] = H\left[\frac{(P + P_e)}{2}\right] - \frac{H[P]}{2} - \frac{H[P_e]}{2} \qquad (6)$$

where $Q_0$ is a normalization constant used to maintain $0 \leq Q \leq 1$. For more details about $Q_0$, please refer to S3 Appendix in S1 File. Note that $Q[P, P_e]$ depends on two different probability distributions. The first one, $P$, is related to the signal under analysis, and the second, $P_e$, is a uniform distribution, which represents a white noise signal. This reference histogram with uniform distribution is considered the equilibrium point of any physical systems.

## The proposed ecoacoustic global complexity index

Given the fundamental concepts in the previous section, we can now summarize the proposed EGCI calculation. For any ecoacoustic signal obtained by a sensor node, these steps must be followed:

1. from a signal $X$, apply the autocorrelation (Eq 1) choosing a maximum value of $\tau_{max}$ to obtain a Toeplitz matrix $R_{xx}$ (Eq 2);

2. apply the Singular Value Decomposition (SVD) on $R_{xx}$ to recover its singular spectrum (*i.e.*, the eigenvalues $\lambda_{1:\tau_{max}}$ of $R_{xx}$); then,

3. normalize each eigenvalue $\lambda_i$ by the sum of all eigenvalues to get the histogram $P$ and calculate the normalized entropy $H[P]$ according to the Eq 3;

4. compute the Jensen-Shannon divergence using the histogram $P$ and a uniform histogram $P_e$ (Eqs 5 and 6); and finally,

5. apply Eq 4 to estimate the EGCI, the complexity index of the ecoacoustic signal.

This procedure maps each signal to a unique point, with $H$ and $C$ coordinates, in the Entropy-Complexity (HxC) plane. Note that, $R_{xx}$ is a full rank matrix, therefore all its eigenvalues, as well as their normalized values, are positive real numbers. This allows us to interpret the singular spectrum as a Probability Mass Function (PMF) of a signal recorded by any acoustics sensor. Moreover, the only free parameter of our method is $\tau_{max}$.

In some specific situations, two or more different signals may have singular spectra with the same entropy. For instance, consider the three points $p_1$, $p_2$, and $p_3$ depicted in Fig 1b. In these cases, divergence plays a key role in helping to separate their complexities. In other words, divergence is useful to separate histograms with equal entropy, as in the Entropy-Complexity plane depicted in Fig 1b. One may note that entropy weighted by the divergence causes a large range of possible $C$ values for each entropy value. Such values are contained between the upper and lower limits, shown in the same figure by the black dashed lines. In this same figure, we

illustrate the distribution of the audio samples from our dataset using color gradients to highlight the effect caused by the Jensen-Shannon divergence term in Eq 4.

Lastly, the dashed blue line in Fig 1 represents the position of simulated color noise signals by a function with Power Spectral Density (PSD) that obeys a power law of the form $\xi(f) = \frac{1}{|f|^\alpha}$ [28, 38], where $f$ denotes frequency. Thus, we vary the alpha parameter between $0 \leq \alpha \leq 2$ in small increments generating several random time series, each of them with a sampling frequency of $f_s$ = 44.1 kHz and a Signal-to-Noise Ratio of SNR = $0kHz$. The position on the HxC-plane of each one of these series draws the blue dashed reference line shown in Fig 1. For more details about colored noises, please refer to S4 Appendix in S1 File.

## Results

In this section, we will analyze the EGCI computed from our experiments in the field.

### Characterization of reference samples

Sueur *et al.* [4] made publicly available seven signal records specifying the acoustic richness of each one according to their index. We can use them as references and investigate the characteristics of these signals by the EGCI. This allows us to validate and compare the proposed index.

Fig 2a shows the complexity of these signals. According to Sueur *et al.*, the recorded signals $s_{19}$ and $s_{18}$ have an elevated biodiversity richness, for their higher entropy values. The EGCI provided deeper information, as we realized that the increase in entropy may be a consequence of both: the cricket's chirping (or other insects) or an increase in environmental noises. Thus, EGCI will not just label those samples with high biodiversity richness, but it is able to provide more information about them. Take, for instance, the following examples.

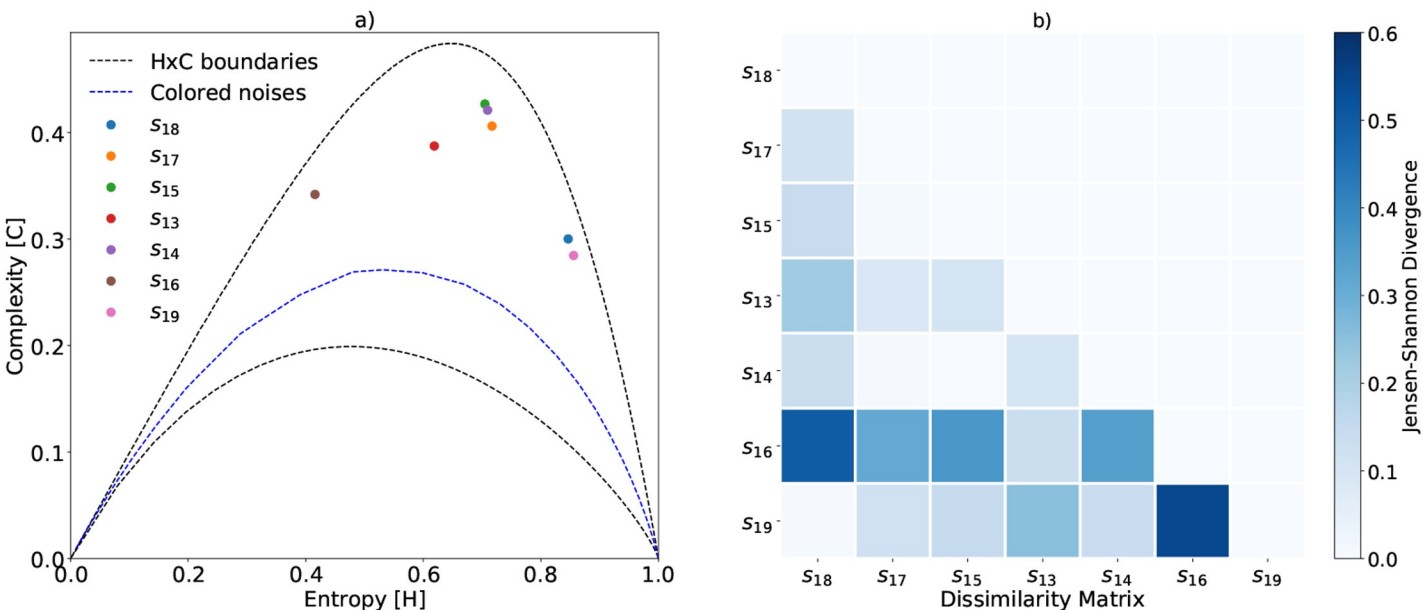

**Fig 2. The characterization of reference signal samples using $\tau_{max}$ = 512.** Every point in subfigure a) represents an ecoacoustic signal. Subfigure b) shows the divergence matrix between signal pairs. We keep the original numbering of the signals to help the interpretation and comparison with Sueur *et al.* [4]. These records are available online at https://doi.org/10.1371/journal.pone.0004065.

By employing the EGCI, we discover that signals $s_{14}$, $s_{15}$ and $s_{17}$ carry a larger number of different deterministic patterns, increasing the complexity, as an indication of higher ecoacoustic richness. The characterization of $s_{13}$ shows that this sample has slightly more noise with few additional vocalization patterns compared to $s_{16}$, but keeping less diversity than, for instance, $s_{14}$. Lastly, $s_{16}$ has the smallest diversity of acoustic patterns, and therefore the smallest complexity and entropy. One has to understand the role of $\tau$ in the HxC plane, and the resulting trade-off when tuning it. A small value of $\tau$ results in a low-discriminant plane, being difficult to visualize the differences among signals. A high value of $\tau$, on the other hand, will split the main frequencies captured by the auto-correlation matrix into several groups.

One of the advantages of using Jensen-Shannon divergence is that we can quantify the difference between two arbitrary signals. Thus, we can make peer-to-peer comparisons between a given signal and a reference signal from another soundscape or between signal variations of the same soundscape at different times. The matrix of Fig 2b shows the divergence between all pairs of reference signals. The colorbar indicates that the greater the divergence between the signals, the stronger the blue color. It is worth noting that there is a correspondence between the distance of the points in the complexity plane and their divergences. For instance, the greatest divergence is obtained by comparing the singular spectra of $s_{16}$ and $s_{19}$, which are the farthest points in the HxC-plane. A similar observation holds when comparing points $s_{16}$ and $s_{18}$. Although the position of the points in this plane uses the uniform distribution as a reference, the dissimilarity between them has a correspondence with the divergences of the signals; that is, the closer the points are in the HxC-plane, the greater is their ecoacoustic similarity.

For the sake of comparison, Fig 3 shows the values of $H_a$, ACI, EGCI and the Von Neumann entropy $H$ used in the proposed methodology. We can see that the $H_a$ has values very close to $H$. This shows that our methodology to compute entropy from eigenvalues captures the same information contained in the signals without the need to calculate $H_t$. We found that $H_t$ values are very close to one in almost all samples in the data set; consequently, the product $H_a = H_f \times H_t$ is not changed and the final value is $H_f$. However, we cannot affirm that this is a general rule. Therefore, as the $H$ captures the same information as the $H_a$, we can say that the EGCI presents complementary information, better weighting the content of the signal in the analyzed samples.

In this figure, we can also notice that the ACI places greater emphasis on the $s_{13}$ sample. The spectral content of $s_{13}$ indicates the presence of several species of animals with intermittent calls in different frequency ranges. Therefore, the derivative of such frequencies with respect to time is high, increasing the ACI. Comparatively, EGCI also assigns a high value to the same sample, but without being the highest value among all. Despite these small discrepancies, the ACI assigns low and almost uniform values to the remaining samples, being less informative in such cases.

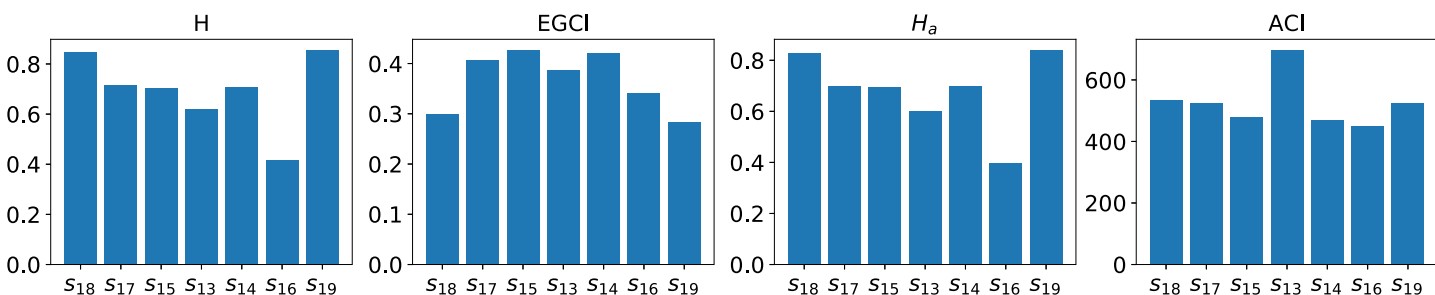

**Fig 3. Values of the three indices, ACI, $H_a$, EGCI, and the entropy of the eigenvalues $H$, for the reference samples.** These values were obtained using a time lag ($\tau$) and an FFT with 512 points.

## Characterization of samples from *in situ* monitoring

Our second investigation used a dataset with signals recorded in the Mamirauá conservation area, in the Brazilian Amazon rainforest. Our acoustic sensor captured approximately 40 Gb of raw audio signals, during thirteen days of the monitoring program. The audios were stored in 22-second segments, with a sampling frequency of $f_s$ = 44.1 kHz, summing up 43348 audio samples. The monitoring program, called *Providence*, was divided into two phases. The former, consisting of five consecutive days of recording in July 2016, and the latter, consisting of eight consecutive days at the beginning of September on the same year. All collected data compose a single data set. Authorization for the use of this sensitive biological data has been released by The Biodiversity Authorization and Information System (SISBio) through number 72722. The raw signals cannot be made available due to legal restrictions. A copy of the original data can be requested through the Providence website http://projectprovidence.org.

Fig 4 depicts the relationship between the main elements of our analysis. The key step of our methodology is how to obtain the probability distributions to calculate entropy. As previously pointed out, the histogram of Eq 3 comes from the normalized eigenvalues of the autocorrelation matrix. Fig 4a shows the complexity plane for $\tau_{max}$ = 512, where the 43348 segments were represented as EGCI points. According to our methodology, points with higher complexity should exhibit a higher acoustic richness. Therefore, to illustrate the behavior of the proposed index, three extreme points were arbitrarily chosen: $s_1$, which has low entropy and low complexity; $s_2$, with medium entropy and high complexity; and $s_3$, with high entropy and complexity lower than $s_2$ but higher than $s_1$. For each of these points, we plot their singular spectrum (Fig 4b), the ACF given by the Eq 1 (Fig 4c) and their PSD spectrograms (Fig 4d). From top to bottom we have $s_1$, $s_2$, and $s_3$, respectively.

We can verify in Fig 4b that ecoacoustic signals with few components and long-range correlations, such as $s_1$, tend to have a concentrated singular spectrum, decreasing $H[P]$, while signals with uncorrelated noises (ie. tending to a white noise), such as $s_3$, have a flat singular spectrum, raising up $H[P]$. In the intermediate case, when the signals have different deterministic patterns, singular spectra similar to that of $s_2$ are generated, causing an intermediate entropy value. The ACF of sample $s_1$ has long-range correlations (Fig 4c), implying an environmental colored noise, and a spectrogram with energy accumulated mainly at low frequencies (top spectrogram in Fig 4d). We verified that this recording corresponds to rain sound without disturbance of animals. As we expected, $s_1$ is very close to the curve of environmental colored noises with $\alpha \approx 2$. Additionally, the ACF of $s_1$ produces the singular spectrum shown in the upper plot of Fig 4b, which justifies its low entropy value.

One can make a similar analysis about $s_2$ and $s_3$. In the case of $s_2$, the ACF plot shows short- and medium-range correlations, with a spectrogram richer in different acoustic patterns (middle plot in Fig 4c). Such ACF produces the singular spectrum shown in the third plot of Fig 4b. The distribution of the eigenvalues of this singular spectrum returns an average entropy value, but its divergence tends to be maximum, thus justifying the high EGCI. From its spectrogram, we also note that there is a low-energy ambient noise spread in a few frequency bands. We verified that this recording has high acoustic richness, containing calls of at least four different bird species, two frog species, and some insects.

Finally, $s_3$ shows high entropy and decreased complexity. Its ACF ($R_{xx}(s_3)$) plot shows only short-range correlations. This may be due to the lack of repeated deterministic patterns and the presence of noises spreading out energy at low, medium, and high frequencies. In this case, the ACF plot has values only for low $\tau_i$ indices, emphasizing the presence of approximately uncorrelated noise, which helps break weak correlations of signal components. As we know, uncorrelated noise signals tend to have an approximated flat singular spectra increasing their

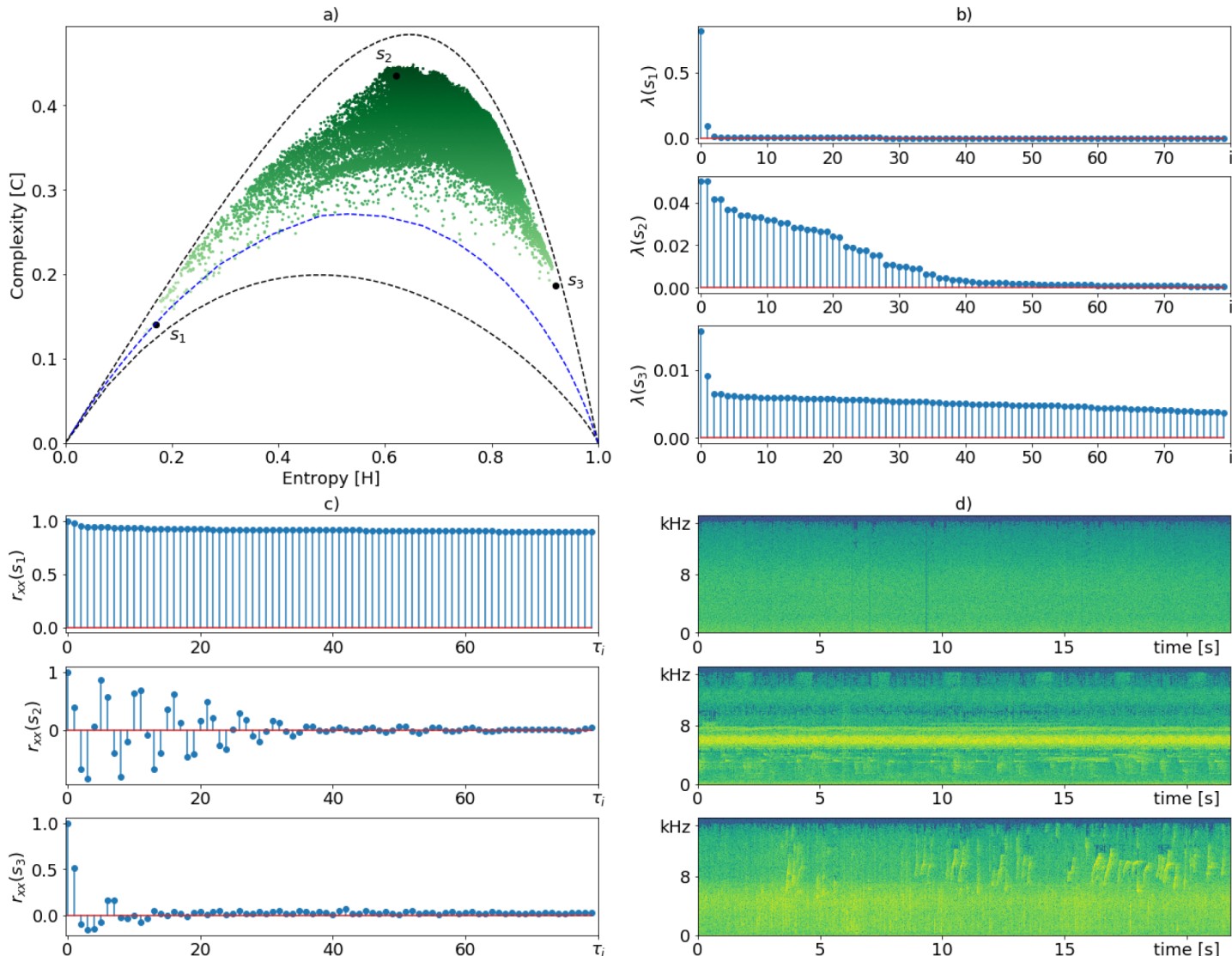

**Fig 4. EGCI characterization using $\tau_{max}$ = 512 of samples recorded at Mamirauá protection area in the Amazon rainforest.** Subfigure (a) depicts the HxC complexity plane, where each point is a signal segment of 22 seconds. Subfigures (b-d) show the singular spectrum, the autocorrelation function (ACF) and the spectrograms, respectively, of points $s_1$, $s_2$, and $s_3$, from top to down.

entropy value, a fact that can be seen in the lower plot of Fig 4b. In the bottom spectrogram of Fig 4d, we observed scattered noises, produced by rain reaching different materials, contaminating most of the frequency bands and also a call of a single species of bird. The presence of this quasi-deterministic pattern prevented the EGCI from achieving even higher entropy and lower complexity.

So far we have discussed how the HxC plane characterization of ecoacoustic samples relates to signal processing concepts, such as ACF and PSD. We also demonstrate, in a practical way, how the proposed index relates to the acoustic richness of a signal segment. The influence of $\tau$ on the distribution of samples in the HxC-plane can be seen in S1 Fig. In the following sections, we present a detailed data analysis on how the proposed index varies in relation to the time of day and across the monitoring period, characterizing temporal soundscapes patterns.

## Temporal characterization through EGCI

The proposed index allows us to characterize the monitored location considering temporal variations of the acoustic richness. Fig 5a shows the complexity of each segment within three half-hour periods in a plane with $\tau_{max}$ = 512. We can verify that there was an entropy increase in the period between 03:00 and 03:30 h (green dots). This may be related to an increase in environmental noises, the intense acoustic activity of insect choral singing, and other environmental factors, as well as a decrease of birds, amphibians, and other animal calls with daytime habits, causing a lower acoustic diversity. The interval corresponding to 12:00 and 12:30 h shows a concentration of blue dots with a slight increase in entropy compared to red dots, mostly due to the combination of the acoustic activity of some animals with daytime habits and insect chorus. The extreme blue dots with low entropy are probably due to the regular rainfall at this time of day during the monitoring period, which decreases entropy. Lastly, we have the interval between 07:00 and 07:30 h (red dots), which presents the highest EGCI values and few dots shifted slightly to the left of the plane—this may be a consequence of higher acoustic activity of birds at dawn.

The complete characterization of variations in acoustic richness at every hour of the day can be better appreciated in Fig 5b. This figure shows the centroids and their respective scatter bars for each sample group at each hour. As we expected, the horizontal bars are always larger than the vertical bars due to the shape of the HxC-plane. Comparing the different groups, we can see which periods of the day present the highest ecoacoustic dispersion (e.g. 12:00 h). Interestingly, at sunset (e.g. 05:00 h) the highest EGCI values are achieved with a lower dispersion, which suggests that there is an intense acoustic activity with less perturbation of environmental phenomena at this time. The second key time is dawn (e.g. 07:00 h) in which it is known that there is a greater acoustic activity of morning birds. Points 21:00, 22:00 and 23:00

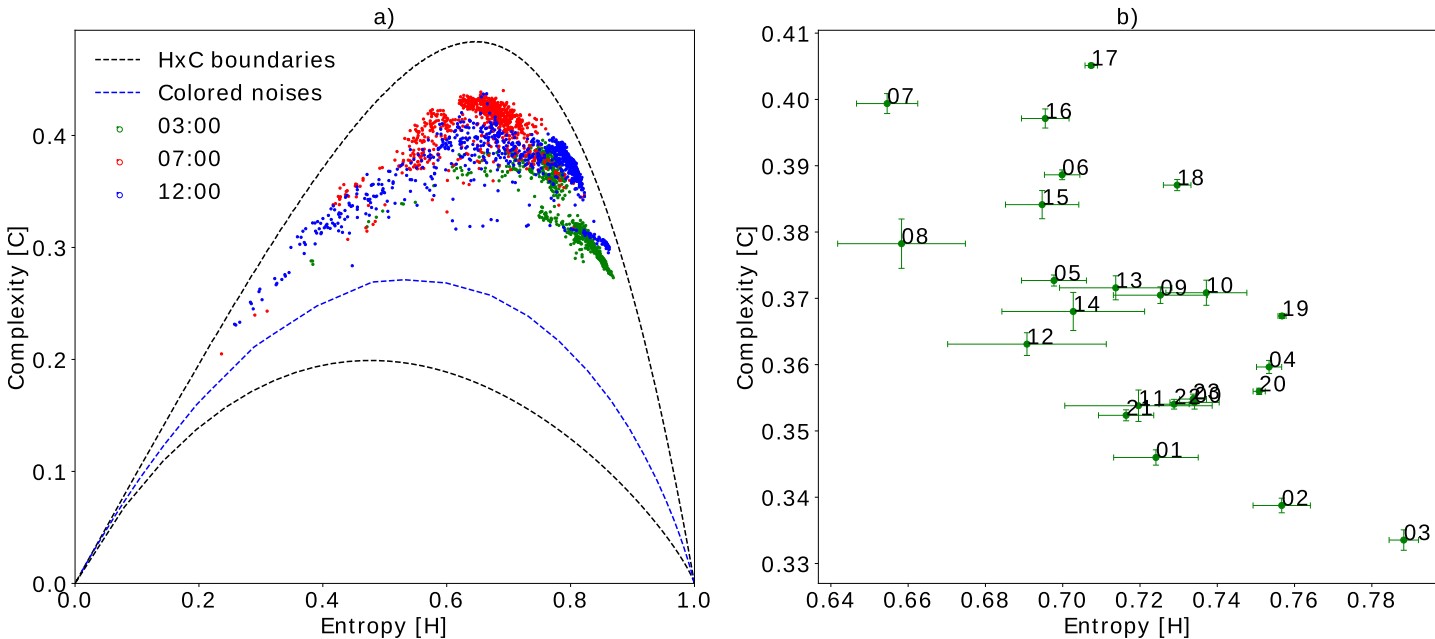

**Fig 5. Temporal grouping of samples using $\tau_{max}$ = 512 during the seven days of monitoring.** Subfigure a) shows the spatial distribution of the samples at three half-hour intervals. Subfigure b) shows the cluster centroids with complexity and entropy variance bars at one-hour intervals. The proximity between these centroids highlight hours of the day with similar ecoacoustic richness.

h are relatively close, and as described in the previous sections, near points in the plane have histograms with low divergence caused by similar acoustic patterns.

Looking at the spatial centroid's distribution in Fig 5b, we also noticed that before dawn (e.g. 03:00 h) the acoustic activity of the birds, anurans and other species is lower, giving rise to greater insect activity such as cicadas, which increases entropy. Finally, we can see groups of centroids with similar characterizations (near location and comparable dispersion), for example, the sets {06:00 to 08:00 h, and 15:00 to 18:00 h} or {05:00, 09:00, 10:00 h, and 12:00 to 14:00 h}. Such centroid variations are better depicted in Fig 6b. This figure presents the EGCI variation with a half-hour resolution, where we can notice two peaks of maximum acoustic activity, at 07:00 and 17:00 h, a fact that matches the knowledge of the experts about soundscapes of the Mamirauá region.

Environmental phenomena may change over the days, however, we expect that this fingerprint of acoustic complexity must be repeated with some frequency, except when biodiversity changes. To verify such consistency through time we included Fig 7. In other words, the daily pattern shown in Fig 6 is almost regular across the six consecutive days of the monitoring program, shown in Fig 7. We observed that the temporal variation of EGCI is little affected by the $\tau$ parameter variation. This parameter increases the scale between the maximum and minimum peaks of acoustic complexity but keeps the shape of the curve. During a few days of the monitoring program, the sensor experienced technical difficulties that caused the loss of some minutes of recording. However, the proposed index proved to be resilient to these issues. The soundscape fingerprint of daily variation including the Entropy and Divergence quantifiers are additional resources presented in S2 Fig.

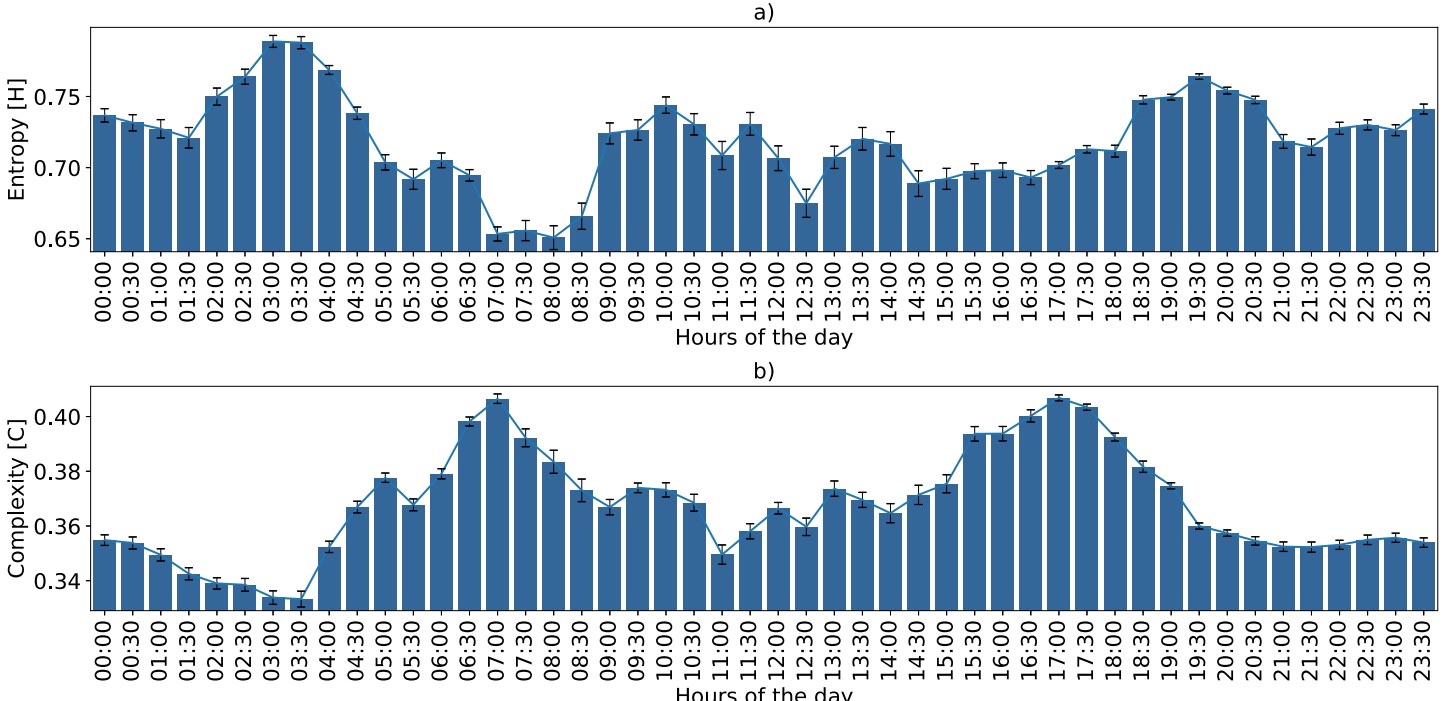

**Fig 6. Ecoacoustic entropy and complexity with its confidence interval at every half hour using $\tau_{max}$ = 512.** Average variation considering the seven days of monitoring. This variation can be considered the soundscape fingerprint of the region within the sensor microphone range.

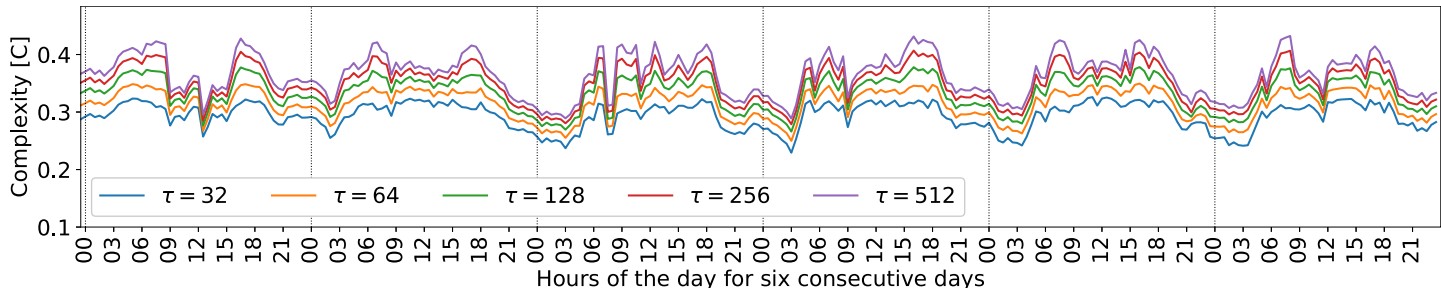

**Fig 7. EGCI regular patterns varying through six days.** Behavior patterns of the EGCI index indicate greater activity during the day and lower activity at night. Daytime variations are more irregular compared to nighttime variations.

### EGCI variability in Mamirauá floodplain during drought and flood seasons

The Mamirauá Sustainable Development Reserve is mostly floodplain, where the different phases of the hydrological cycle influence the ecosystem [39, 40]. In Mamirauá, the seasons are strongly marked by the water level. During the flood season the land is completely underwater, while during the dry season, there is plenty of dry land available. These changes impose mobility restrictions for some species, which directly modifies the ecoacoustic landscape. As mentioned at the beginning, the monitoring period was divided into two months, approximately one week in July and another week in September during flood and drought event peaks, as shown in S3 Fig. These two months are characterized by the flood season and the dry season, directly affecting the landscape. Stratifying our dataset by month and plotting the EGCI for each week, we observed a shift over the spatial distribution of points in Fig 8). The effect of this seasonal variation can best be observed by the centroid displacement of each month. This comparison shows the usefulness of EGCI in capturing the variation of the ecoacoustic soundscape due to the change of season.

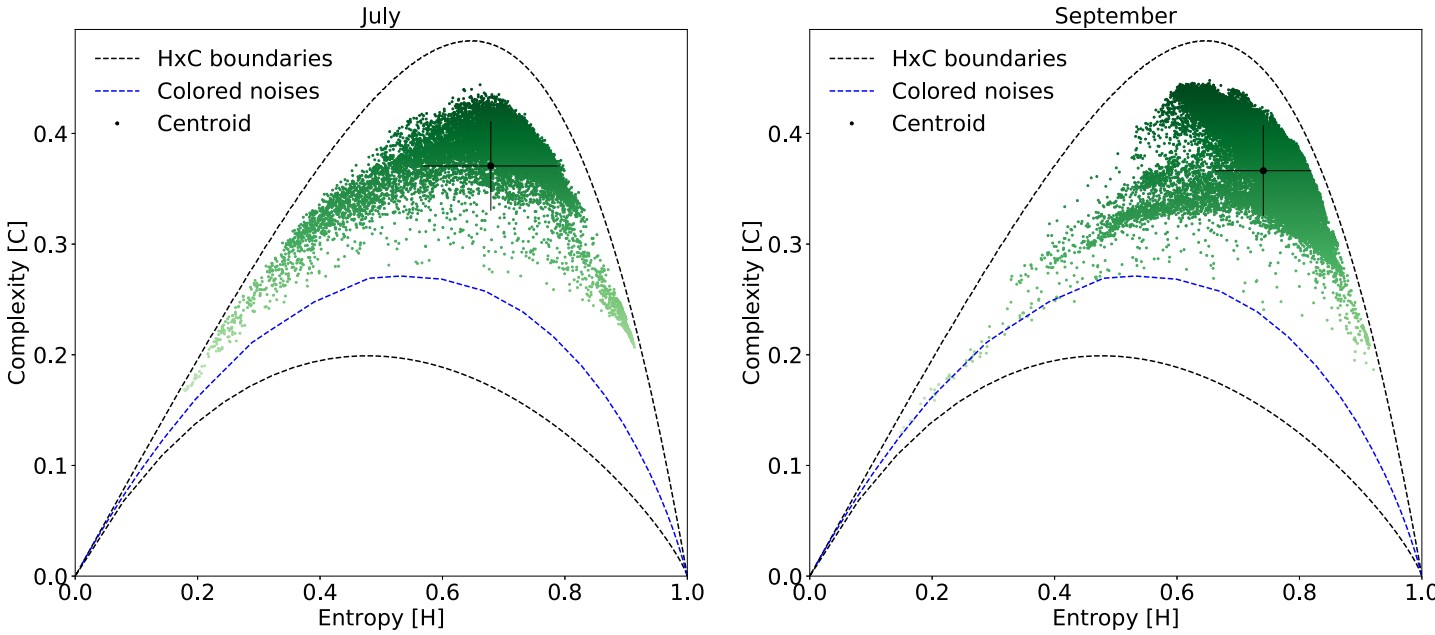

**Fig 8. Seasonal characterization of the soundscape using tau = 512 $\tau_{max}$.** This possibility of characterization can be extended to long term monitoring programs.

## Rainfall sound characterization

Environmental phenomena are also captured by the sensor's microphone. Consequently, environmental sounds can also be characterized in the HxC-plane. We know that these phenomena—for instance, the sound of the rain, have a frequency spectrum known as colored noises. This is the main reason why we included the simulation of this type of noise, generating the blue reference curve in Fig 9a.

This specific monitoring period was within the dry season [41]. However, on September 2, 2016, there was a slight amount of rainfall during the interval between 11:00 and 11:30 h. Only the acoustic samples corresponding to this day are illustrated in the HxC plane of Fig 9a. Here, the red dots correspond to a half-hour interval starting at 11:00 h. During this interval there was rainfall, and the resulting approximation of the samples to the color noise curve. In this same time interval, both entropy and complexity decreased abruptly, as can be seen in Fig 9b. The left shift of the red dots in the HxC-plane causes a considerable increase in the variance of the information quantifiers, as indicated by the confidence interval on the red bar. After the rain stopped, the entropy quickly recovered to a high value, while the complexity still took time to increase. For those with field experience, it can be seen that after a storm the forest is silent for a few minutes, recovering acoustic activity first by insects, which increases entropy, and then with birds and other animal species, ultimately increasing ecoacoustic complexity. In order to verify this, the recorded audios were hand-inspected by a specialist, who heard and confirmed rain sound at the mentioned time period.

These recordings were also analyzed using the ACI and $H_a$ indices S4 Fig was added in supplementary material for comparison purposes. The $H$ highlighted the rain phenomenon better than the $H_a$ index, indicating that the methodology presented for calculating entropy is more discriminatory (Fig 9b vs S4a Fig). Comparing the complexity obtained by the ACI with the EGCI, we noticed that this last index better characterized the periods before and after the rain. The ACI attributed a low value of complexity to rain, but comparable to other values, such as in the periods from 3:00 to 04:00 h or 20:00 to 00:00 h, making it difficult to discriminate against this environmental phenomenon (Fig 9c vs S4b Fig). However, discriminating against environmental phenomena is a subjective task that requires supervised inspection; therefore, additional experiments must be carried out to reach a more comprehensive conclusion.

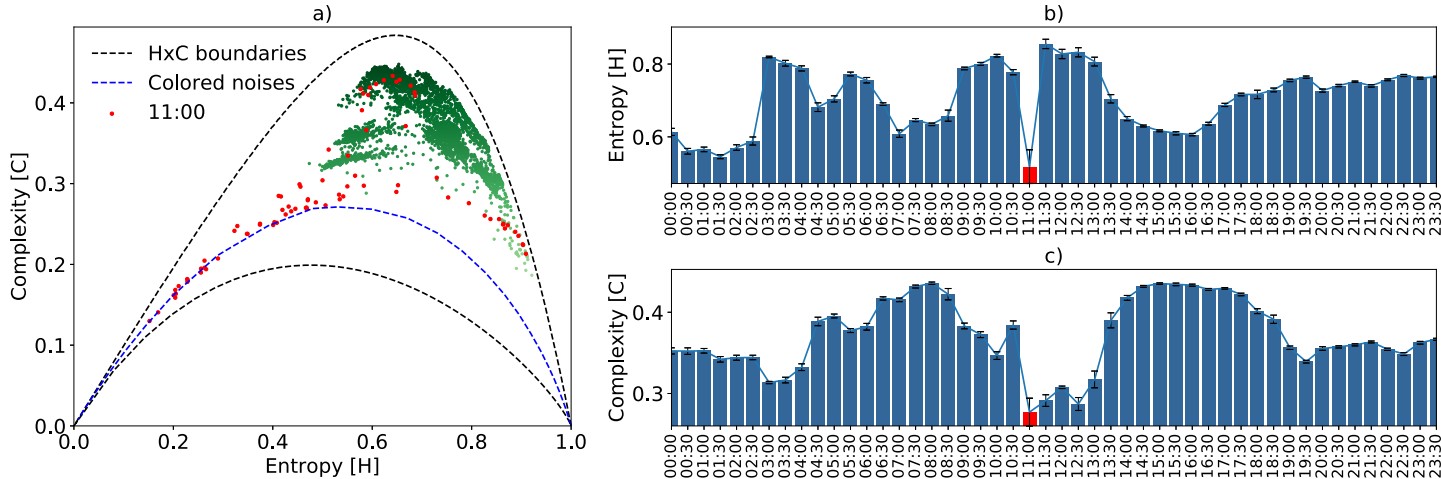

**Fig 9. Rainfall characterization using $\tau_{max}$.** Subfigure a) shows the EGCI from September 2, 2016. The red dots correspond to a half hour interval when it rained. Subfigure b) shows the average variation, along with their respective confidence intervals, of the quantifiers during that day every half hour.

## Discussion

In previous sections, we presented the theoretical bases and empirical evidence that support the composite index of ecoacoustic complexity. By nature, ecoacoustic signals are a mixture of deterministic and stochastic components. Therefore, our main challenge was to make the EGCI index sensitive to bioacoustic diversity while differentiating among environmental phenomena. Thus, the physical interpretations of the HxC plane contribute to understanding of the soundscape characterizing the monitored area.

The ability to represent different patterns using autocorrelation depends on the parameter $\tau$. Adjusting this parameter is a trade-off between separability and representation in our methodology. Small values of $\tau$ limit the ability to discriminate all acoustic patterns possibly contained in the signal, and very high values of $\tau$ cause the acoustic patterns to decompose into smaller segments that must remain together, causing an artificial increase in entropy. On the other hand, large $\tau$ values allow a more discriminating HxC plane, giving more space to the spatial characterization of points, while small $\tau$ values generate a plane with closer upper and lower limits, reducing the ability to discriminate between signals with different degrees of stochasticity and determinism (S1 Fig).

Through the results presented in the previous sections, whether using reference or real data, we show that the EGCI correlates well with the diversity in the recordings. In the HxC-plane, vocalizations of species that have regular spectral-temporal patterns balance entropy, whereas species with irregular calls—with a greater number of different patterns—tend to increase entropy. We also observed that insect choral singing, which generally varies between 5 kHz and 8 kHz, like cicadas, increases entropy because they have a spectrogram similar to an uncorrelated high-frequency noise. As the complexity is a concave function, the increase in entropy caused by insects of the same species generally decreases the complexity indicating less diversity, regardless of how many individuals the coral has. If they are alone and belong to the same species, then the diversity is smaller and consequently, the complexity should decrease (Fig 5a). In this sense, EGCI is able to adequately characterize the presence of insect choruses in the HxC plane region of high entropy.

Short- and long-term environmental phenomena are also well characterized by EGCI. Short-term sporadic phenomena, as verified by the example of the rain, can be recognized with our approach. In this example, the low-frequency rain sound—characterized as colored noises—decreases entropy and complexity, whenever not disturbed by other sounds, generating HxC points close to the colored noises reference curve (Fig 9). With regard to the long-term environmental phenomena, such as the hydrological cycle, there is also a different characterization of samples in the HxC plane for different seasons. Mamirauá is a floodplain area with a season of the year completely flooded. This change in landscape alters the dynamics of the species and hence, the soundscape, as was characterized in Fig 8.

Additionally, the EGCI was able to highlight temporal regularities consistent with the observations made in the field, discriminating against any irregularities in the soundscape that occurred during the monitored days (Fig 7). Besides, EGCI can naturally quantify the dissimilarity between two recordings, as this index was defined in terms of the Jensen-Shannon divergence (Fig 2b). Compared to other indices, we show that the EGCI characterizes the Mamirauá region in a more appropriate way than the ACI. The methodology presented to obtain entropy $H$ captures similar dynamics of those captured by $H_a$ more directly, while adding information about the divergence. Therefore, the advantage of EGCI is that it can be interpreted as a metric of general complexity and, additionally, each of its terms can be used as indicators of the soundscape variation, since they have physical interpretations.

Regarding our analysis with Mamirauá records, the temporal analysis presented above shows that there is a regularity of the soundscape between hours of the day and across monitored days. As expected, from Figs 6 and 7 we noticed a higher daytime acoustic activity. Fig 6b highlights two high EGCI peaks at dawn and dusk, an indication of the hours of the day with higher intensity and acoustic diversity, which is a typical behavior of birds and frogs in the Amazon rainforest. As shown in Fig 7, this behavior is almost regular between consecutive days. It was also possible to notice that there is a change in the characterization of the soundscape correlated with flood and drought seasons (Fig 8), which indicates a change in the species present during those seasons.

As pointed out by Fuller *et. al.* [25], not every index is adequate to represent all possible landscape configurations through the analysis of soundscapes. This may be due to temporal variations in the acoustic patterns of animal communities, which are strongly linked to the local conditions of natural phenomena. From this perspective, in all of our temporal analyzes, for example, comparing night vs day time or flood vs drought seasons, there are indications that EGCI captures significant changes related to time and also to the patterns of environmental phenomena affecting the Mamirauá region. However, it is worth mentioning that the conclusions reached correspond to the soundscape of the region covered by the range of the microphone used by the sensor node and that our recordings do not contain anthropogenic noises, only geophonic noises.

## Conclusion

Information Theory provides powerful and elegant methods for ecoacoustic signal analysis, such as the Entropy-Complexity plane theory to map generalized statistical complexity from a particular soundscape. Therefore, we are not referring only to the dynamics of a specific species, but to the dynamics of the environment as a whole. To compose the EGCI, we combined the *Von Neumann* entropy, calculated from the eigenvalues of the autocorrelation matrix, with the *Statistical Complexity*. It is worth mentioning that this new index encapsulates the representation of any ecoacoustic signal as a two-dimensional point in the Entropy-Complexity (HxC) plane. In addition to its low-dimensional representation, each point on the HxC plane has useful interpretations regarding the underlying physical system. Therefore, the EGCI characterizes nontrivial sound correlations and can be applied to ecoacoustic signals with variable temporal length, only requiring the adjustment of a single parameter ($\tau$).

In addition to these contributions, we detailed the dynamics of the HxC plane, its upper and lower bounds, the effect of divergence on the spatial characterization of the samples, the effect of the $\tau$ autocorrelation parameter, and also add a curve simulating the colored noises possibly found in nature. These features allow the differentiation of signals where natural patterns, such as the singing of birds, amphibians, insects or other animals, and environmental phenomena are combined. Moreover, the low computational complexity of the SVD algorithm used to obtain the eigenvalues makes the method interesting for contexts with limited hardware resources. Additionally, we provide the source code for experiment replication together with the figures script at https://bit.ly/EGCI_index, facilitating the reproduction of results and possible comparisons with other methods.

The usefulness of EGCI was assessed using a real data set from *Mamirauá Reserve*. In this remote, hard-to-reach location, biodiversity remains virtually unchanged. Furthermore, the region is under the influence of the flood cycle of the Amazon river, reaching its maximum typically in June. The available records are from July (water level is high) and September (water level is close to the minimum), representing the two characteristic seasons of this region

well. These recordings allow us to obtain a baseline to compare with acoustic landscapes modified by human intervention in future studies.

A direct application of EGCI is the characterization of periodic seasons. As shown in the Mamirauá Reserve case study, seasonal variations of the water level are the most important factor in determining the presence of different species communities present in flooded areas in the Amazon. The proposed index proved sensitive enough to capture the soundscape variations due to the change from flood season to drought. In future investigations, we would like to expand our data analysis to characterize the acoustic dynamics of other regions and also perform a more in-depth comparison with all the indices mentioned in the literature review.

## Supporting information

**S1 Audio.**
(FLAC)

**S2 Audio.**
(FLAC)

**S3 Audio.**
(FLAC)

**S1 Fig. The relationship between the $\tau$ parameter and the HxC plane.** The separation between the upper and lower limits makes it possible to better discern the distribution of points in the HxC plane. We notice, in S1. Fig, how the spatial distribution of points changes in relation to $\tau$. The higher the $\tau$, the greater the distance between the lower and upper boundaries, allowing new grouping patterns to appear.
(EPS)

**S2 Fig. Radar chart plot of 24h.** Radar chart plot using $\tau_{max} = 512$. From left to right the three information quantifiers used in our methodology. Each chart in S2 Fig shows the quantifier's variation over the hours of the day. These charts can be adopted as the fingerprint of the monitored soundscape.
(EPS)

**S3 Fig. Hydrological cycle of Mamirauá floodplain.** S3 Fig shows the water level phases according to the months of the year 2016, measured in meters above sea level (MASL or m.a.s.l.). This data is publicly accessible through the link https://www.mamiraua.org.br/fluviometrico-na-reserva.
(EPS)

**S4 Fig. Rainfall characterization using FFT = 512.** Subfigure a) shows the $H_a$ and subfigure b) shows the ACI from September 2, 2016. The bars quantify the average variation together with their respective confidence intervals.
(EPS)

**S1 File.**
(PDF)

## Acknowledgments

The authors would like to thank the Instituto de Desenvolvimento Sustentável Mamirauá for their support in coordinating the Providence Project. The authors thank the Gordon and Betty Moore Foundation for their grant support and scholarships. We are grateful to all the

institutions that took part in the Providence Project, including the Commonwealth Scientific and Industrial Research Organization (CSIRO), The Sense of Silence Foundation and the Technical University of Catalonia. We also thank the Coordenação de Aperfeiçoamento de Pessoal de Nível Superior (CAPES), for the institutional support, and the Fundação de Amparo à Pesquisa do Estado do Amazonas (FAPEAM)—PAPAC Project (Public Notice 005/2019).

## Author Contributions

**Conceptualization:** Juan G. Colonna.

**Investigation:** Juan G. Colonna, José R. H. Carvalho.

**Methodology:** Juan G. Colonna, Osvaldo A. Rosso.

**Project administration:** José R. H. Carvalho.

**Software:** Juan G. Colonna.

**Supervision:** José R. H. Carvalho.

**Validation:** José R. H. Carvalho, Osvaldo A. Rosso.

**Visualization:** Juan G. Colonna.

**Writing – original draft:** Juan G. Colonna.

**Writing – review & editing:** José R. H. Carvalho, Osvaldo A. Rosso.

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
