## [Decision Letter · Decision Letter 0]

17 Apr 2020

PONE-D-20-02986

The Amazon rainforest soundscape characterized through Information Theory quantifiers

PLOS ONE

Dear Dr. Colonna,

Thank you for submitting your manuscript to PLOS ONE. After careful consideration, we feel that it has merit but does not fully meet PLOS ONE’s publication criteria as it currently stands. Therefore, we invite you to submit a revised version of the manuscript that addresses the points raised during the review process.

Both reviewers were complementary of the overall approach in the paper and both were supportive of this metric getting published but both also had numerous issues with the writing in the manuscript itself so please address all the comments from both reviewers.  I would also note that the emdedded hyperlink did not work for myself or either reviewer so please correct that as well.

We would appreciate receiving your revised manuscript by Jun 01 2020 11:59PM. To enhance the reproducibility of your results, we recommend that if applicable you deposit your laboratory protocols in protocols.io, where a protocol can be assigned its own identifier (DOI) such that it can be cited independently in the future. For instructions see: http://journals.plos.org/plosone/s/submission-guidelines#loc-laboratory-protocols

We look forward to receiving your revised manuscript.

Kind regards,

Dennis M. Higgs

Academic Editor

PLOS ONE

Journal Requirements:

Reviewers' comments:

Reviewer's Responses to Questions

**Comments to the Author**

1. Is the manuscript technically sound, and do the data support the conclusions?

Reviewer #1: Yes

Reviewer #2: Yes

2. Has the statistical analysis been performed appropriately and rigorously? 

Reviewer #1: N/A

Reviewer #2: Yes

3. Have the authors made all data underlying the findings in their manuscript fully available?

Reviewer #1: No

Reviewer #2: Yes

4. Is the manuscript presented in an intelligible fashion and written in standard English?

Reviewer #1: No

Reviewer #2: Yes

5. Review Comments to the Author

Reviewer #1: The title of the paper could be more informative

Very interesting paper, but are the authors aware that the name of the index is too close to ACI index? This could create confusion, not competition. I suggest to change the name of the index to better differentiate it for the future literature (when possible).

The result section is too long and is very hard to follow it. I suggest to reshape and shorten when possible. The reader has difficulties to follow it in some parts.

In the discussion a better and more extended comparison with the other two indices would be welcome.

Line 2: Environmental better than ecological ?

Line 6: soundscape’s biodiversity seems to me not appropriate terminology

Line 17: species richness, …., and insects? Seems to me not well linked

Line 28: not only diversity but also dynamics, daily and seasonal cycles, etc. please rephrase

Line 30: I recommend to add an intermediate sentence explaining the necessity to propose a new index

Line 38: frequency better than spectral?

Line 32 to 64: It seems to me more a conclusion or an extended summary and not an introduction. I suggest to split in the methods and in part in the concluding comments. The content is ok but it seems to me that does not belong to an introduction. For instance, at line 32 “The proposed Ecoacoustic Complexity Index.. “ But this is the first time that the index is cited…

Line 64: http://bit.ly/2m12PWc seems not to work

Line 78: please add some references

Line 101-104: The author say things that are not correct. For instance at line 103 there are no indices that can work after a hardware malfunction!!

Line 104, the index has been proposed to evaluate the level of complexity inside a soundscape, or if you prefer it measures the amount of acoustic information present at every frequency bin.

Line 109: should assumed? Or should assume

Line 117: Not to defend ACI but ACI produces an acoustic signature really different for white noise or heavy rains, or dawn chorus. Second, is absolutely wrong to say that ACI depends on the length of the recorded files, if you average the total ACI….

Line 121-122: Please modify to be consistent with the previous comments

Line 410: I don’t understand, please rephrase

Line 412: This is not a new result, but is confirmed by ACI

Line 415: Also this is common to other well established metrics, where is the novelty? Please explain better

Line 416: Please explain the position of the insects

Line 418-423. I am sorry, but this section is really confused and contradictory please rephrase

Line 424: Again this is not a peculiarity of ECI but other indices have found these patterns

Line 404: In this discussion it seems a confusion between what the ECI is able to marks and the character of the study areas. For clarity the authors should separate in the discussion these two parts that now are stirred.

Line 448: I don’t believe that this is an important point

Line 462: the http does not work?

Line 468: ecoacoustics landscape? Why not soundscape?

Line 469-471: Looks not necessary, your paper has methodological characters and references to other biomes in Brazil contaminates the “universal” valence of the ECI.

Reviewer #2: In this manuscript the authors describe a new ecoacoustic index and demonstrate how linking entropy and complexity over space and time adds value to our understanding of a soundscape. The impact of the work could be enhanced with better framing in the introduction (see below) and a clearer contrast between the ECI and other acoustic indices. Addressing these would help readers who are users of acoustic indices see the value of the contribution.

Line 19 - The first two sentences are both true, but not logically linked. The recent paper (citation #1) only came out last year, thus the observations have not driven the behavior in part two. A simple solution would be to flip these, with the second a narrower application of monitoring animal populations. ** however after reading the whole paper, this seems not to be the focus...

Line 22 – There are far more than two ways to monitor change in animal populations! Visual surveys, camera traps, genetics sampling… I see where the authors are going, and it is a good direction, but the opening paragraph needs to be more inclusive of the broader research (at least as currently phrased)

Line 72 – I would clarify the differences between the supervised and unsupervised methods. In particular, what the focus of each tool is in the field. Is it to describe the biological community or change in a population or is it to described the soundscape and how it might change. These are two complementary but different goals.

Line 105 – how are you defining acoustic richness? Does it compare to species richness in your application? Indeed clarity in what biological vs acoustic measures of focus and the relationship between them should be addressed

Line 228 – but the data from Saeur is said about to have the highest richness a s19 and s18. Is ecoacustic richness different then biodiversity richness? How are these applicable as measures of animal populations overtime (from the opening paragraph)?

Line 267 – this seems like the most useful interpretation. Perhaps in fig 1 use this term, complexity plane, and describe and interpret it for readers coming from the use/application side to see how it adds value to other measures.

Perhaps also show how the ECI compares to the indexes mentioned above (and others noted below) to show how the additional information from the ECI adds value.

1. Normalized Difference Soundscape Index (NDSI) is the proportion of biophony to anthrophony in the soundscape, calculated as (biophony - anthrophony) / (biophony + anthrophony) (Kasten et al. 2012);

The Acoustic Diversity Index (ADI; Villanueva-Rivera et al. 2011) measures the diversity of sounds; it is the proportion of signals in each bin above a threshold, with the final value calculated with the Shannon diversity index;

2. The Acoustic Evenness Index (AEI) measures the equality/inequality of distribution of sound power in different frequency ranges (Villanueva-Rivera et al. 2011) and is calculated in the same way as the Acoustic Diversity Index, but with the Gini index of evenness; and

3. The Bioacoustic Index (BAI) is a function of both the power and frequency range of sound generated by wildlife; it is calculated as the area under each curve, including all frequency bands associated with the dB value that was greater than the minimum dB value for each curve (Boelman et al. 2007).

6. PLOS authors have the option to publish the peer review history of their article (what does this mean?). If published, this will include your full peer review and any attached files.

Reviewer #1: No

Reviewer #2: No

---

## [Author Response · Author response to Decision Letter 0]

5 Jun 2020

Dear editor and reviewers, we appreciate your valuable comments that helped us improve the quality of our research. All requested changes have been made.

The responses to the reviewers' comments were answered and attached as a single document called "Response to Reviewers.pdf", included in the list of documents sent through the system.

The response to comments made by the editor was included at the end of the Cover Letter. Therefore, the cover letter has also been updated. The new version was loaded into the list of documents requested by the system.

In case of any doubt, please feel free to contact us at anytime.

Sincerely,

Juan, 

Reginaldo, 

Osvaldo.

---

## [Decision Letter · Decision Letter 1]

29 Jun 2020

Quantifying ecoacoustic activity in the Amazon rainforest through Information Theory quantifiers

PONE-D-20-02986R1

Dear Dr. Colonna,

We’re pleased to inform you that your manuscript has been judged scientifically suitable for publication and will be formally accepted for publication once it meets all outstanding technical requirements. Only one of the original reviewers was able to review the revision but they felt satisfied that all the revision were properly carried out, and after my own review I agree. Thank you for your care in the revisions.

Kind regards,

Dennis M. Higgs

Academic Editor

PLOS ONE

Additional Editor Comments (optional):

Reviewers' comments:

Reviewer's Responses to Questions

**Comments to the Author**

1. If the authors have adequately addressed your comments raised in a previous round of review and you feel that this manuscript is now acceptable for publication, you may indicate that here to bypass the “Comments to the Author” section, enter your conflict of interest statement in the “Confidential to Editor” section, and submit your "Accept" recommendation.

Reviewer #2: All comments have been addressed

2. Is the manuscript technically sound, and do the data support the conclusions?

Reviewer #2: Yes

3. Has the statistical analysis been performed appropriately and rigorously? 

Reviewer #2: Yes

4. Have the authors made all data underlying the findings in their manuscript fully available?

Reviewer #2: Yes

5. Is the manuscript presented in an intelligible fashion and written in standard English?

Reviewer #2: Yes

6. Review Comments to the Author

Reviewer #2: I greatly appreciate the edits made. The contrast to past work is stronger and the language more clear.

One last suggestion- add the text between the "" in line 2

These two observations have driven many researchers to monitor the variations of animal populations through time "using acoustic measures and indicators", and to use them as indicators of 5

environmental degradation

7. PLOS authors have the option to publish the peer review history of their article (what does this mean?). If published, this will include your full peer review and any attached files.

Reviewer #2: No

---

## [Editor Report · Acceptance letter]

7 Jul 2020

PONE-D-20-02986R1 

Quantifying ecoacoustic activity in the Amazon rainforest through Information Theory quantifiers 

Dear Dr. Colonna:

I'm pleased to inform you that your manuscript has been deemed suitable for publication in PLOS ONE. Congratulations! Your manuscript is now with our production department. 

Kind regards, 

on behalf of

Dr. Dennis M. Higgs 

Academic Editor

PLOS ONE